# Dietary Habits and Choices of 4-to-6-Year-Olds: Do Children Have a Preference for Sweet Taste?

**DOI:** 10.3390/children8090774

**Published:** 2021-09-02

**Authors:** Malgorzata Kostecka, Joanna Kostecka-Jarecka, Mariola Kowal, Izabella Jackowska

**Affiliations:** 1Department of Chemistry, Faculty of Food Science and Biotechnology, University of Life Sciences in Lublin, 20-950 Lublin, Poland; izabella.jackowska@up.lublin.pl; 2Independent Public Healthcare Center in Łęczna, 21-010 Łęczna, Poland; kostecka-joanna@wp.pl; 3Cardinal Stefan Wyszyński Specialist Regional Hospital, 20-718 Lublin, Poland; mariolakowal13@gmail.com

**Keywords:** food preference, sweet taste, children, kindergarten, dietary habits

## Abstract

Children develop food preferences by coming into direct contact with various food products through the senses of taste, touch, sight and smell. The aim of this study was to analyze the food preferences of children aged 4 to 6 years and to determine whether age and gender influence children’s food preferences and whether the preference for sweet taste changes with age. The study involved a paper questionnaire containing images of 115 different food products and dishes. The respondents expressed their preferences by choosing the appropriate emoji (happy, sad or neutral face). The study was conducted between 2018 and 2020, and it involved 684 children from 10 kindergartens. Girls chose a significantly higher number of foods and dishes they liked than boys (*p* = 0.002), and 4-year-olds gave a higher number of “neutral” responses than 5- and 6-year-olds (*p* = 0.001). Dietary diversity increased with age, and younger children were familiar with fewer foods than 6-year-olds (*p* = 0.002). Children had a clear preference for sweet taste, regardless of age and gender. Young children (4-year-olds) were more likely to accept healthy foods despite the fact that they were familiar with fewer products and dishes.

## 1. Introduction

Childhood is a period of rapid growth and development. In this critical phase, food preferences are formed and carried into childhood and beyond, and foundations are laid for a healthy adult life [1]. Early life exposures may contribute to the risk of obesity [2,3,4]; therefore, the eating habits of young children are recognized as a topic of great social and public health interest [5,6].

The diversity of children’s diets is a very important research problem because only a well-balanced and varied diet can guarantee the supply of all necessary nutrients. A knowledge of children’s familiarity with different food groups and the factors that shape children’s food preferences can contribute to the prevention of diet-related diseases [7,8]. Children become more familiar with foods upon repeated exposure, and their acceptance of new foods increases over time. Previous analyses have supported the hypothesis that early exposure to new tastes and textures influences children’s food preferences and acceptance of new foods in later life, and that factors such as gender and age may play a role in children’s food preferences [9,10].

Taste preference (TP) is one of the factors that affect children’s food intake and eating habits [11]. The sweet taste is associated with sugar and its derivatives, such as fructose or lactose, but other substances in fruit juices or drinks can also activate sensory cells that respond to sweetness. Vennerød et al. [12] explored the taste sensitivity of children aged 4 to 6 years and observed that sensitivity to sweetness decreased with age. In another study analyzing children’s preference for intense sweetness, more children than adults selected the “super happy” face emoji on a 5-point scale [13]. Experimental research has shown that heightened preference for sweet taste is also evident in the real life setting and that children consume a greater percentage of calories from added sugars [14,15]. The consumption of sugars, in particular sugar-sweetened beverages, by European children and adolescents exceeds current recommendations [16]. Added sugars in daily foods are responsible for around 14% of the energy intake in the diets of 2- to 9-year-olds in Europe [17] and 2- to 18-year-olds in the USA [18,19]. Sweet taste preferences vary across geographical regions even in Europe and are probably related to the weight status of European children [20]. In studies conducted on the Polish population, only several attempts have been made to determine whether preschool children’s preferences for sweet taste increase with age [21,22,23], and most of the conducted research focused on the food preferences and dietary patterns of schoolchildren.

Food preferences, especially for sweet and fatty taste, and the consumption of sugar-sweetened beverages (SSBs) could be strong risk factors for increased weight and/or obesity in children [24,25]. Children who consumed SSBs during infancy had higher odds of obesity at 6 years than non-SSB consumers [26]. The role of early nutritional experience and its impact on the risk of overweight and obesity in children has been well documented [27,28]. Further research is needed to identify the factors that predispose children to develop unhealthy eating habits and a preference for sweet taste which increase the risk of obesity and other diseases [29]. Liem and de Graaf have shown that exposure to sweetened orange soda for 8 days increased the preference for sweet taste in 9-year-olds, but not in adults [30]. It remains unclear whether this effect remains stable over time and whether it can be extrapolated to other sugar-rich foods. A prospective study of 166 girls in the USA demonstrated that children who drank soda at the age of 5 years had higher mean consumption of soda at 7 to 15 years of age [31]. The aim of this study was to analyze the food preferences of children aged 4 to 6 years, to evaluate the diversity of children’s diets, children’s familiarity with different food groups, and to determine whether children have a preference for sweet-tasting foods. The research hypothesis postulates that age (6-year-olds vs. 4-year-olds) and gender influence food preferences and that older children are more likely to select sweet tasting products.

## 2. Materials and Methods

### 2.1. Methods

The study involved a paper questionnaire containing images of 115 different food products and dishes that are most commonly served to children in Polish kindergartens and homes. The pictures depicted food products classified into 12 food groups based on WHO and FAO guidelines [32], as well as ready meals and processed foods that are available in Poland. The foods and beverages included in the analysis were chosen based on the factor analyses conducted by Jilani et al. [33] who categorized foods into the respective taste modalities. The following product groups were presented in the questionnaire: vegetables, fruit, milk and dairy products, bread and cereal products, meat, fish, processed meat, eggs, legumes, beverages, sugar, sweets and pastry, soups, dishes served for lunch/dinner, and fast foods. The questionnaire was administered in paper form. Foods and dishes were depicted in large color photographs to enable all children to easily recognize the presented products. The respondents used pencils and felt-tip pens to mark their preference for each product / meal. Children expressed their preferences for specific foods and dishes by choosing the appropriate emoji (happy, sad or neutral face). The research tool was developed in collaboration with a pre-school education expert, and it had been previously applied in studies of preschoolers and elementary school students [21,34] (Figure 1).

The diversity of children’s diets was assessed based on the number of products/meals that were known to the respondents in each of the 12 groups. The sad or neutral face emoji denoted products or groups of products that were not familiar to the respondents or were not included in their diets. The happy face emoji denoted products that were known to and consumed by children.

### 2.2. Study Design and Participants

The questionnaires were completed by children at kindergarten or during counseling sessions in a nutrition clinic. The time of questionnaire administration was not limited; children could ask questions, and they marked the answers independently. Parental approvals were obtained for the surveys conducted in kindergartens, or parents attended counselling sessions in the nutrition clinic with their children, but they did not fill in the questionnaire. The study was conducted between October 2018 and January 2020, and it involved 712 children from 10 kindergartens participating in the “Health-Promoting Kindergarten” program in Lublin (south-eastern Poland) and 88 children who attended counselling sessions in the nutrition clinic with their parents. The returned questionnaires were evaluated for completeness, and 684 correctly filled questionnaires were used in the analysis.

The inclusion criteria were: age of 4–6 years, absence of diseases that require an elimination diet (such as celiac disease, food allergy or phenylketonuria), and the child’s ability to complete the questionnaire (assessed by the kindergarten teacher). The presented results are part of a long-term research study entitled “Eating Colorful, Eating Healthy”, which has been conducted since 2018 in Polish kindergartens participating in the “Health-Promoting Kindergarten and School” nation-wide program. The study also involved parent surveys, dissemination of knowledge about healthy nutrition in kindergartens, and a repeated survey of children. The results will be processed and presented in subsequent papers.

### 2.3. Data Analysis

Categorical variables were presented as a sample percentage (%). Differences between groups were verified with the chi^2^ test (categorical variables) or the Mann-Whitney test (continuous variables). Before statistical analysis, the normality of variable distribution was checked in the Kolmogorov-Smirnov test. For continuous variables, data are presented as means with a 95% confidence interval (95% CI—confidence intervals). The significance of ORs (odds ratios) was assessed by Wald’s statistics. In the first part of the study, age and gender were used as variables to test the respondents’ preferences for different food products. In the second stage, the analyzed foods and dishes were divided into three groups of products based on the main sensory attribute, i.e., sweet, fatty and bitter taste. The presence of relationships between age and food preferences was analyzed. All analyses were conducted in the Statistica 12.0 PL (Poland) program (StatSoft Inc., Tulsa, OK, USA, StatSoft, Krakow, Poland). The results were regarded as statistically significant at *p* < 0.05.

## 3. Results

The study involved 256 boys (37.4%) and 428 girls (62.6%). Four-year-olds accounted for 19.6% (*n* = 134) and 6-year-olds–for 52.8% (*n* = 361) of the studied population. Most children had a preference for sweets and pastry, fruit and dairy products, and significant differences were not observed between genders (*p* > 0.05) or age groups (*p* > 0.05). The most disliked foods were cereal products such as oatmeal and coarse groats, leafy vegetables such as cabbage and dairy products such as plain yogurt or kefir (Table 1). Girls liked significantly more foods and dishes than boys (*p* = 0.002), and 4-year-olds gave a higher number of “neutral” responses than 5- and 6-year-olds (*p* = 0.001). Legumes, meat and cereal products such as groats and rice were most frequently selected as “neutral” foods.

Dietary diversity increased with age, and younger children were familiar with fewer foods than 6-year-olds (*p* = 0.002). Age was also a factor in the children’s food preferences (Table 2). Four-year-olds had a greater preference for natural and relatively unprocessed foods, such as milk, fermented dairy products, fresh vegetables and fruit, plain breakfast cereals and high-quality cured meats, whereas older respondents were familiar with a higher number of foods and dishes (*p* = 0.001) and had a greater preference for processed foods, including fast foods (*p* = 0.001), sweet snacks and sweetened beverages (*p* < 0.05).

In comparison with 6-year-olds, 4-year-olds were less likely to choose products such as hamburgers (OR= 0.56; Cl: 0.39–0.89, *p* = 0.006), hot dogs (OR = 0.73; Cl: 0.59–0.94, *p* = 0.003), potato chips (OR = 0.71; Cl: 0.63–0.90, *p* = 0.0002) and French fries (OR = 0.84; Cl: 0.76–1.04, *p* = 0.009) as the foods they liked. Age did not differentiate the respondents’ preferences for pizza and tortillas (*p* > 0.05). Younger children significantly less often selected sweetened hot beverages (OR = 0.79; Cl: 0.68–1.04, *p* = 0.008), carbonated soft drinks (OR = 0.66; Cl: 0.51–0.88, *p* = 0.002) and flavored water (OR = 0.81; Cl: 0.72–1.03, *p* = 0.002) as the preferred beverages, whereas juice was liked equally in all age groups (*p* > 0.05).

One of the research objectives was to determine whether children have a preference for sweet/sweetened products with high sugar content. Regardless of gender, 5- and 6-year-olds had a significantly greater preference (*p* < 0.05) for sweetened dairy products such as yogurt, sweet cream cheese and flavored milk than for milk and plain fermented dairy products. All children liked sweets and sweet snacks. Kinder chocolate, milk chocolate, cookies, lollipops, ice-cream and chewing gum were selected as the preferred products by both girls and boys of all ages. Only gummy candy and candy bars were more often selected by boys (OR = 1.41; CI: 1.17–1.84, *p* = 0.005) and 6-year-olds (OR = 1.24; CI: 1.09–1.36, *p* = 0.02).

An analysis of the dishes served for lunch/dinner revealed that children had a greater preference for meatless sweet dishes, such as pancakes and crêpes, than salty dishes. A similar trend was observed in the respondents’ choice of beverages, dairy products and pastry such as Danish pastries.

In the next stage of the analysis, the evaluated foods were divided into three groups of products with a sweet, fatty and bitter taste. Foods and meals were classified into each group based on the main sensory attribute (mostly taste) and composition that was responsible for the sweet (fruit, sweetened beverages, sweetened dairy products, sweets), fatty (meat, fat, processed foods such as fast foods, Nutella, nuts) and bitter taste (black tea, spices, endive, lettuce, rocket and eggplants). The presence of correlations between the respondents’ age and preference for sweet, fatty and bitter taste was determined (Table 3).

Dark chocolate and black tea were also classified as bitter-tasting foods, but they were not liked by children in any age group. Fatty-tasting foods, including fast foods, fried and grilled foods, were less often liked by 4-year-olds (OR = 0.79; CI: 0.54–0.93, *p* = 0.02), and were most often liked by 6-year-olds (OR = 1.47; CI: 1.12–1.63, *p* = 0.003).

## 4. Discussion

Children’s food preferences and dietary habits are influenced mainly by environmental and social factors. Children have a natural preference for sweet taste and high-energy foods abundant in fat and simple carbohydrates [13,35]. The above does not imply that such foods should be incorporated in children’s diets. Exposure to healthier and less calorie-dense foods that are less abundant in simple sugars can effectively modify children’s food preferences. Early experiences with nutritious foods and flavor variety increase the likelihood that children will choose a healthier diet in later life because they like the taste and the variety of healthy foods [35,36,37]. Parents’ dietary behaviors and kindergarten meal programs also play a very important role in teaching children to make healthy nutritional choices and explore new flavors and foods [38,39]. Research has shown that exposure to products/dishes with high nutritional quality can be effective in expanding the diet, introducing new flavors and shaping healthy nutritional choices. A knowledge of children’s dietary preferences is needed to introduce beneficial changes in both the family and the preschool environment [40,41,42,43,44].

Age is a factor that influences children’s development and active exploration of the world, including food products. At first, children do not make their own food choices but rely on their parents/caregivers to provide them with the appropriate foods. By the age of 3 or 4 years, children develop autonomous feeding behavior and set boundaries on the foods they will accept [45,46,47]. Children learn to explore foods, experience new tastes, flavors and products, and they gradually begin to make independent dietary choices that are initially limited by the choice of foods at home, and subsequently, in the school cafeteria or the supermarket [45,46]. In this context, the results of this study confirm the theory that older children are familiar with more foods.

Gender is also an important factor that shapes healthy eating habits from an early age. Studies conducted on teenagers demonstrated that girls chose healthier foods and snacks, and tended to eat products that were less processed and calorie-dense, such as vegetables [47,48,49], fish and dairy products [50,51]. Wardle found that girls made healthier food choices and consumed more fruit and vegetables than boys already at the age of 4 [52]. Similar observations were made in the present study.

The present study revealed that 4- to 6-year-olds have a clear preference for sweet-tasting foods, are familiar with a larger number of sweet-tasting foods and dishes, and are less likely to dislike these items. According to Mennella [53], Lanfer [20] and Sobek [24], children prefer sweetened products such as breakfast cereals, juice, soft drinks, sweets and puddings. The present study also demonstrated that children enjoyed fruit which is naturally sweet, and that their preference for fatty taste, in particular fast foods, increased with age. Similar observations were made in the population of European children and teenagers [11,37], as well as in a review article [54]. Children’s preference for high-fat products is explained by association with the energy provided by fat.

### Strengths and Limitations

The main strength of this study was the large number of children who completed the questionnaire. Most studies investigating the dietary preferences of kindergarten children rely on questionnaires that are filled by parents and caretakers. In this study, the respondents made their own choices by selecting the appropriate emojis. This is a unique approach in research exploring the dietary preferences and habits of children who have not yet learned to read or write. The results indicate that children can readily match concepts of food products and dishes to emotional correlates (like, don’t like, don’t know/neutral) before they develop literacy skills. This is the first Polish study conducted on a large group of respondents to demonstrate that literacy is not an absolute requirement for children to identify foods.

The reliability of the results could be a certain limitation due a general scarcity of studies involving children who have not yet learned to read or write. However, the pictorial questionnaire had been successfully used in a previous study, and the results were consistent with the answers provided by the parents. A similar study was conducted on 3-year-olds by Privitera [55].

## 5. Conclusions

The results of the present study confirmed the hypothesis that children’s food preferences change with age, mostly in favor of less healthy foods. Regardless of age and gender, the surveyed children had a clear preference for sweet taste, which disproves the hypothesis that older children are more likely to select sweet-tasting foods than younger children. This observation could be useful in shaping healthy dietary habits and preferences in this age group. Despite the fact that young children were familiar with fewer products and dishes than older respondents, they were more likely to accept healthy foods, in particular plain dairy products and vegetables, and their preference for processed foods with a fatty taste increased with age. The results of this study provide valuable insights for developing nutrition education curricula in kindergartens and planning meals that expose children to products that are less liked, but deliver health benefits.

## Figures and Tables

**Figure 1 children-08-00774-f001:**
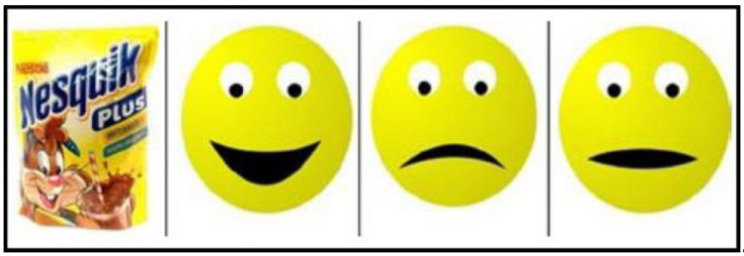
A screenshot of a food item [21].

**Table 1 children-08-00774-t001:** Examples of products/dishes that were most liked and disliked by the respondents.

Product/Dish	Most Liked	Most Disliked
Vegetables	Tomatoes, chives, corn, cucumbers, carrots	Radishes, cabbage, broccoli, pumpkins, onions
Fruit	Apples, oranges, tangerines, kiwis, strawberries, bananas, watermelons	Grapefruit, black and red currants, plums, melon
Dairy products	Fruit yogurt, sweet cream cheese, milk, plain cream cheese, yellow cheese	Plain yogurt, kefir, cream
Meat, fish	Poultry ham, frankfurters, pâté, sausage	Cured meat with visible fat, such as gammon or bacon
Cereal products	Wheat rolls, wheat bread, whole-grain rolls, extruded puff corn snacks, chocolate cereals	Wholemeal bread, oatmeal, buckwheat groats, rice
Legumes	Green peas	Dried peas, chickpeas
Nuts, seeds, grains	Peanuts, cashews, sunflower seeds	Walnuts, pumpkin seeds
Sweets	Kinder chocolate, milk chocolate, gummy candy, fruit ice-cream, Nutella	-
Fast food	Pizza, tortillas, French fries, potato chips	Kebabs, hamburgers
Dishes served for lunch/dinner	Fried and breaded meat, dumplings, crêpes, spaghetti	Fried meat patties, fried chicken, potatoes
Beverages	Juice, sweetened black tea, water, homemade fruit drinks	Cereal coffee, herbal tea such as mint tea

**Table 2 children-08-00774-t002:** Odds ratio (95% confidence interval). The association between liked foods and dishes and the respondents’ age and gender.

Product	Boys (Ref. Girls)	4-Year-Olds (Ref. 6-Year-Olds)	5-Year-Olds (Ref. 6-Year-Olds)
Vegetables	0.78 * (0.66–1.09)	1.19 * (0.98–1.23)	1.03 (0.87–1.21)
Citrus fruit	1.07 (0.81–1.24)	1.04 (0.92–1.29)	1.10 (1.01–1.25)
Fruit with small seeds	1.09 (0.95–1.14)	0.94 (0.78–1.06)	0.97 (0.71–1.06)
Apples, pears and other pome fruit	1.23 * (1.09–1.32)	1.07 (0.89–1.13)	1.02 (0.89–1.18)
Milk	0.81 * (0.74–1.06)	1.19 * (1.03–1.34)	1.05 (0.95–1.34)
Sweetened dairy products	1.06 (0.89–1.12)	1.07 (0.93–1.24)	0.97 (0.82–1.19)
Plain fermented dairy products	1.03 (0.87–1.11)	1.28 * (1.06–1.44)	1.09 (0.89–1.21)
Wheat bread	0.99 (0.78–1.23)	1.09 (1.01–1.34)	1.26 * (1.03–1.41)
Wholemeal bread	0.79 * (0.64–0.99)	1.11 (0.95–1.26)	1.08 (0.93–1.21)
Plain breakfast cereal	1.03 (0.78–1.11)	1.23 * (1.01–1.34)	1.07 (0.95–1.27)
Sweetened breakfast cereal	1.08 (0.89–1.18)	1.12 (0.99–1.31)	0.95 (0.76–1.27)
Pastry	1.21 * (1.04–1.46)	1.08 (1.03–1.36)	1.07 (1.01–1.13)
High-quality cured meats, such as ham	0.74 * (0.64–0.97)	0.74 * (0.61–0.96)	1.05 (0.86–1.26)
Processed meats such as frankfurters and pâté	1.08 (1.02–1.29)	1.10 (1.02–1.28)	1.31 * (1.09–1.47)
Mineral water	1.24 * (1.13–1.36)	1.26 * (0.88–1.39)	1.11 (0.96–1.24)
Flavored water	1.07 (0.96–1.23)	0.81 * (0.72–1.03)	1.28 * (1.13–1.49)
Fruit juice/sweetened fruit nectar	1.19 * (0.89–1.48)	1.04 (0.87–1.14)	0.98 (0.87–1.12)
Sweetened hot beverages	0.96 (0.78–1.17)	0.79 * (0.68–1.04)	1.05 (0.81–1/23)
Unsweetened hot beverages	1.07 (0.96–1.20)	1.29 * (1.16–1.47)	0.98 (0.84–1.21)
Chocolate/candy bars	1.02 (0.89–1.06)	1.06 (0.96–1.21)	0.97 (0.84–1.13)
Cookies	0.97 (0.78–1.14)	1.05 (1.01–1.09)	1.07 (0.91–1.17)
Gummy candy	1.07 (0.89–1.19)	0.88 * (0.67–0.98)	1.03 (0.78–1.27)
Lollipops/chewing gum	1.23 * (1.04–1.36)	1.06 (1.03–1.09)	1.04 (0.91–1.26)
Ice-cream	1.07 (0.96–1.14)	1.03 (0.87–1.24)	1.03 (0.84–1.26)

Statistically significant: * *p* < 0.05.

**Table 3 children-08-00774-t003:** Odds ratio (95% confidence interval). The association between age and taste preferences of preschool children.

	Sweet Taste			Fatty Taste			Bitter Taste		
	OR	95% CI		OR	95% CI		OR	95% CI	
Age (ref. all children)	1.00			1.00			1.00		
4-year-olds	1.04	0.89	1.12	0.79 *	0.54	0.93	1.21 *	0.84	1.37
5-year-olds	1.07	1.02	1.15	1.13	1.03	1.24	0.80 *	0.71	0.97
6-year-olds	1.12	0.98	1.26	1.47 **	1.12	1.63	1.09	0.84	1.23

Statistically significant: * *p* < 0.05; ** *p* < 0.01.

## Data Availability

The data presented in this manuscript are available on request from the corresponding author. The data are not publicly available.

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
