# Peer review of "Dietary Habits and Choices of 4-to-6-Year-Olds: Do Children Have a Preference for Sweet Taste?"

_children, 2021, doi:10.3390/children8090774_

Round 1

Reviewer 1 Report

The authors describe food preferences of 684 children aged 4 to 6 years old. The sample size was large, allowing for comparisons between genders and age groups. The questionnaire that assessed food preferences contained 115 food products, providing rich data on a wide range of foods. Although there are study strengths, I recommend major revision addressing the following major and minor concerns:

Major concerns:
-The authors do not provide rationale for the study aims in the Introduction. Why analyze the food preferences of children aged 4 to 6 years old? Why evaluate the diversity of children’s diets? Why evaluate children’s familiarity with different food groups? Why determine whether children have a preference for sweet-tasting foods? Why determine whether age and gender influence children’s food preferences and whether the preference for sweet foods changes with age? In the Introduction, the authors instead provide information on how children develop food preferences (e.g., repeated exposures) and information on taste preferences more generally (e.g., humans can perceive 6 tastes). Neither of these information areas explain to the reader why it is important to investigate the study questions.  It will be important for the authors to majorly revise the Introduction to provide rationale. For example, to provide rationale for testing whether children have a preference for sweet-tasting foods, the authors may want to provide information on how preference for sweet-tasting foods may confer risk for certain health outcomes. To provide rationale for testing whether age and gender influence children's food preference, the authors may want to provide information on how age may determine how many foods children have been exposed to, and gender may influence parental feeding behaviors. 

-The authors do not provide a thorough literature review of the the prior literature on food preferences in children, diversity of children's diets, children's familiarity with different food groups, and food preferences by age and gender. The authors do provide review of one study in Poland on preferences for sweet foods in children as they age. It will be important for the authors to majorly revise the Introduction to provide a stronger literature review. Given the universality of certain food preferences (e.g., preference for sweet foods), the authors should consider expanding their literature review to non-Polish populations. 

-The authors do not provide information on how children's familiarity with different food groups was measured in the Methods section. 

-The authors do not provide information on how "diversity of children's diets" was measured and operationalized in the Methods section. 

-The Results section would benefit from revision/re-organization by directly linking results with the study aims. The authors should refrain from discussing the results (e.g., "Children also liked fruit which is naturally sweet and caters to the demand for the sweet taste which is the preferred taste already in childhood" and "Dark green vegetables, leafy vegetables (endive, iceberg lettuce and rocket) and spices (dill seeds) have a bitter taste") in the Results section and reserve that information for the Discussion section. 

-The authors present information in the Discussion section that is not clear to the reader why it is relevant to the current study. For example, the authors discuss social learning mechanisms and parents'/caretakers' behaviors but these factors were not examined in the current study. It will be important for the authors to revise Discussion to focus on current study. 

Minor comments:
-Since only 684 children completed questionnaires, please refer to this as the sample size number in the Abstract (not "800 children"). 

-The last sentence of the abstract (i.e., "Despite the fact that....") has typos in it and is unclear.

-Please move the following information about the questionnaire from the Results to the Methods section: "The following product groups were presented in the questionnaire: vegetables, fruit, milk and dairy products, bread and cereal products, meat, fish, processed meat, eggs, legumes, beverages, sugar, sweets and pastry, soups, dishes served for lunch/dinner, and fast foods." 

-In Table 1, chives are both in "most liked" and "most disliked" category. How is that possible?

-Please use the terminology "food preferences" throughout rather than "taste preferences." The measure used in the study was a food rating questionnaire not a taste test, so the terminology "food preferences" seems more appropriate. 

-The authors mention in the Discussion that, "The reliability of the results could be a certain limitation, but the pictorial questionnaire had been used in a previous study, and the results were consistent with the answers provided by the parents." Please describe in more detail. 

-Please review the manuscript for minor English issues (e.g., incomplete sentence structures). 

Author Response

Dear Reviewer,

Thank you for revising our manuscript entitled ‘Dietary habits and choices of 4- to 6-year-olds: do children have a preference for sweet taste?’.

We greatly appreciate the time and efforts to review our manuscript and we agree that the proposed changes will contribute to the improvement of our manuscript. We have addressed all issues indicated in reviews, and we believe that the revised version can meet the journal publication requirements.

Please find our responses to the Reviewers’ comments attached. The changes made in the text are highlighted in red.

Yours Sincerely,

Malgorzata Kostecka

Thank you very much for your insightful review. We greatly appreciate the time and efforts to review our manuscript and we agree that the proposed changes will contribute to the improvement of our manuscript. We hope you will find our improvements appropriate and comprehensive.

line

It will be important for the authors to majorly revise the Introduction to provide rationale. For example, to provide rationale for testing whether children have a preference for sweet-tasting foods, the authors may want to provide information on how preference for sweet-tasting foods may confer risk for certain health outcomes. To provide rationale for testing whether age and gender influence children's food preference, the authors may want to provide information on how age may determine how many foods children have been exposed to, and gender may influence parental feeding behaviors.

Thank you very much for this attention. Indeed, the construction of the Introduction was not completely consistent with the content discussed in the further part of the thesis. Thanks to the very valuable remark of the reviewer, the chapter has been modified. Some of the content has been removed, we have described in more detail the topic of sweet taste preferences, the impact of taste / food preferences on health, and especially the possibility of developing excess weight. However, there are no studies on the influence of parents' gender on the shaping of children's taste / food preferences, and this was also not the topic of our work. We have shown that this topic is discussed more often in foreign literature, and there is little research in the population of Polish children. Hence the interest in this topic.

38-43

48-53

60-63

The authors do not provide a thorough literature review of the the prior literature on food preferences in children, diversity of children's diets, children's familiarity with different food groups, and food preferences by age and gender. The authors do provide review of one study in Poland on preferences for sweet foods in children as they age. It will be important for the authors to majorly revise the Introduction to provide a stronger literature review. Given the universality of certain food preferences (e.g., preference for sweet foods), the authors should consider expanding their literature review to non-Polish populations. 

thank you very much for this suggestion. Due to the fact that the Introduction layout was different, the proposed topics were not discussed too extensively. However, after re-editing this chapter, we have described in more detail the preferences for sweet taste in research around the world and the impact / consequences of this on children's health. Indeed, there is a lack of such studies in the Polish population, and if they are, they mainly concern school-age children or teenagers. Therefore, our research may be important for learning about the preferences of Polish preschoolers.

52-57

61-73

The authors do not provide information on how children's familiarity with different food groups was measured in the Methods section. 

information has been completed

87-94

The authors do not provide information on how "diversity of children's diets" was measured and operationalized in the Methods section. 

information has been completed

99-102

The Results section would benefit from revision/re-organization by directly linking results with the study aims. The authors should refrain from discussing the results (e.g., "Children also liked fruit which is naturally sweet and caters to the demand for the sweet taste which is the preferred taste already in childhood" and "Dark green vegetables, leafy vegetables (endive, iceberg lettuce and rocket) and spices (dill seeds) have a bitter taste") in the Results section and reserve that information for the Discussion section. 

Thank you very much for this suggestion, excerpts have been removed or partially moved to the discussion

222-223

The authors present information in the Discussion section that is not clear to the reader why it is relevant to the current study. For example, the authors discuss social learning mechanisms and parents'/caretakers' behaviors but these factors were not examined in the current study. It will be important for the authors to revise Discussion to focus on current study. 

Thank you very much for this suggestion, the excerpts on social learning mechanisms have been removed, while the influence of parents on shaping taste preferences in children is very important, it has been rewritten. Knowing the problems can be important in planning the education of parents and children.

198-202

Since only 684 children completed questionnaires, please refer to this as the sample size number in the Abstract (not "800 children"). 

Thank you very much for this suggestion, the notation has been corrected

20

The last sentence of the abstract (i.e., "Despite the fact that....") has typos in it and is unclear.

this paragraph has been redrafted

24-26

Please move the following information about the questionnaire from the Results to the Methods section: 

Thank you very much for this suggestion, the notation has been corrected

87-90

In Table 1, chives are both in "most liked" and "most disliked" category. How is that possible?

thank you for your attention, there was indeed an error. Among vegetables "I don't like" should be onion.

Please use the terminology "food preferences" throughout rather than "taste preferences." The measure used in the study was a food rating questionnaire not a taste test, so the terminology "food preferences" seems more appropriate. 

thank you very much for this attention, it has been taken into account and corrections have been made

The authors mention in the Discussion that, "The reliability of the results could be a certain limitation, but the pictorial questionnaire had been used in a previous study, and the results were consistent with the answers provided by the parents." Please describe in more detail. 

this paragraph has been redrafted

the authors' point was that perhaps a graphic questionnaire filled in by non-literate children could be considered unreliable. But the questionnaire used has already been used successfully in earlier studies and in studies by other authors in a group of even younger children.

238-239

Please review the manuscript for minor English issues (e.g., incomplete sentence structures). 

the manuscript was read and corrected by a native speaker.

In response to your concerns over the quality of the English language used in our manuscript, we would like to clarify that since the authors are not native speakers of English, the manuscript has been translated by a professional translator who has extensive experience in editing scientific manuscripts. It has also been spell-checked and grammar-checked by a native English speaker. The manuscript has been revised once gain to increase the clarity of presentation. Grammar, spelling and style have been checked. We would be grateful if you could consider the revised manuscript for publication. Should you have any further remarks concerning the use of English in the paper, we would very much appreciate it if you could provide us with specific examples of errors that should be corrected.

Reviewer 2 Report

The methodology as well as study design are not good. Basic methodological informations are not presented.

My concerns are as follows:

  1. Whether the study was approved by Ethic Committee? This should be clearly stated in methodology, together with institutional review board reference or in special part of temple (in the end).  The study should have the approval of the bioethics committee.
  2. Authors use wrog tample - without logotyp of Children. Good one authors can find: https://www.mdpi.com/files/word-templates/children-template.dot
  3. The limitations as stated by the authors are reasonable and can be appreciated and considered.

    4. Overall, the authors have analyzed the available data to a reasonable conclusion.

Author Response

Dear Reviewer,

Thank you for revising our manuscript entitled ‘Dietary habits and choices of 4- to 6-year-olds: do children have a preference for sweet taste?’.

We greatly appreciate the time and efforts to review our manuscript and we agree that the proposed changes will contribute to the improvement of our manuscript. We have addressed all issues indicated in reviews, and we believe that the revised version can meet the journal publication requirements.

Please find our responses to the Reviewers’ comments attached. The changes made in the text are highlighted in red.

Yours Sincerely,

Malgorzata Kostecka

Thank you very much for your insightful review. We greatly appreciate the time and efforts to review our manuscript and we agree that the proposed changes will contribute to the improvement of our manuscript. We hope you will find our improvements appropriate and comprehensive.

The methodology as well as study design are not good. Basic methodological informations are not presented.

Thank you very much for this remark, indeed the methodology has been described too generally, therefore the subsection has been supplemented.

87-94

99-102

Whether the study was approved by Ethic Committee? This should be clearly stated in methodology, together with institutional review board reference or in special part of temple (in the end).  The study should have the approval of the bioethics committee.

As we stated in the publication, the conducted study did not require the approval of the Bioethics Committee. The study was an anonymous questionnaire, no data obtained allowed for the identification of the mother or the child, so in accordance with the guidelines, we did not ask for approval from the commission to conduct this study.

Authors use wrog tample - without logotyp of Children. Good one authors can find: https://www.mdpi.com/files/word-templates/children-template.dot

Thank you for paying attention. We have indeed used a form for another MDPI magazine for which we apologize and are making corrections.

The limitations as stated by the authors are reasonable and can be appreciated and considered.

Thank you very much for this statement. Describing the strengths and weaknesses of the manuscript is very important as it can be used in planning further research.

Overall, the authors have analyzed the available data to a reasonable conclusion.

Thank you very much for this statement.

Round 2

Reviewer 1 Report

The authors adequately addressed most reviewer comments. However, I still have concerns regarding the rationale for the study in the Introduction:

-I recommend deleting the second paragraph of the Introduction. The authors have not made clear why this information is relevant to the current study. The authors are not testing how children develop food preferences, relations of the family environment with child food preferences, and how repeated exposures affect food preferences. Including this information is confusing for the reader.

-I appreciate the additions to the review of prior literature on sweet taste preferences among children; however, the current study aim was not only to examine if children have a preference for sweet-tasting foods. Why evaluate the diversity of children’s diets? Why evaluate children’s familiarity with different food groups? Why determine whether age and gender influence children’s food preferences? The answers to these questions are still not addressed in the Introduction. 

-Please start a new paragraph at the sentence, "Food preferences, especially for sweet and fatty taste, and the consumption of sugar-sweetened beverages (SSBs) could be strong risk factors for increased weight and/or obesity in children [24-25]."

Author Response

Dear Reviewer,

Thank you for revising our manuscript entitled ‘Dietary habits and choices of 4- to 6-year-olds: do children have a preference for sweet taste?’.

We greatly appreciate the time and efforts to review our manuscript and we agree that the proposed changes will contribute to the improvement of our manuscript.

The changes made in the text are highlighted in blue.

Yours Sincerely,

Malgorzata Kostecka

Thank you very much for your insightful review. We greatly appreciate the time and efforts to review our manuscript and we agree that the proposed changes will contribute to the improvement of our manuscript. We hope you will find our improvements appropriate and comprehensive.

line

I recommend deleting the second paragraph of the Introduction. The authors have not made clear why this information is relevant to the current study. The authors are not testing how children develop food preferences, relations of the family environment with child food preferences, and how repeated exposures affect food preferences. Including this information is confusing for the reader.

as suggested by the Reviewer, the amendment was introduced, the second paragraph was deleted

I appreciate the additions to the review of prior literature on sweet taste preferences among children; however, the current study aim was not only to examine if children have a preference for sweet-tasting foods. Why evaluate the diversity of children’s diets? Why evaluate children’s familiarity with different food groups? Why determine whether age and gender influence children’s food preferences? The answers to these questions are still not addressed in the Introduction. 

The second paragraph has been rewritten, taking into account the issues proposed by the Reviewer. Thank you for this suggestion, indeed such a layout will be more suited to the content discussed in the research part.

Literature was also added, while the items from the deleted paragraph were moved to the discussion (they were already cited there).

34-43

291-305

378-382

Please start a new paragraph at the sentence, "Food preferences, especially for sweet and fatty taste, and the consumption of sugar-sweetened beverages (SSBs) could be strong risk factors for increased weight and/or obesity in children [24-25].

thank you for this suggestion, the comment has been included in the text

62

Reviewer 2 Report

Dear Authors!

In temple on the end you can find information:

"Institutional Review Board Statement: In this section, please add the Institutional Review Board Statement and approval number for studies involving humans or animals. Please note that the Editorial Office might ask you for further information. Please add “The study was conducted according to the guidelines of the Declaration of Helsinki, and approved by the Institutional Review Board (or Ethics Committee) of NAME OF INSTITUTE (protocol code XXX and date of approval).” OR “Ethical review and approval were waived for this study, due to REASON (please provide a detailed justification).” OR “Not applicable.” for studies not involving humans or animals. You might also choose to exclude this statement if the study did not involve humans or animals."

https://www.mdpi.com/files/word-templates/children-template.dot

The authors couldn't delate this point. It is a place where you should explain why you needn't approve (if Your law allows it).

Author Response

Dear Reviewer,

Thank you for revising our manuscript entitled ‘Dietary habits and choices of 4- to 6-year-olds: do children have a preference for sweet taste?’.

We greatly appreciate the time and efforts to review our manuscript. We have added the Institutional Review Board Statement paragraph as is in line with the Children's Journal guidelines. The consent of the bioethics committee is not required in the case of anonymous questionnaire studies on child nutrition, without the analysis of laboratory tests or the possibility of identifying the respondent.

The changes made in the text are highlighted in blue.

Yours Sincerely,

Malgorzata Kostecka

Round 3

Reviewer 2 Report

Thank you for your work.

Author Response

Dear Reviewer,

Thank you very much for your insightful review. We greatly appreciate the time and efforts to review our manuscript and we agree that the proposed changes will contribute to the improvement of our manuscript. 

with best regards

Malgorzata Kostecka and Authors